Distribution of phylogenetic groups, adhesin genes, biofilm formation, and antimicrobial resistance of uropathogenic Escherichia coli isolated from hospitalized patients in Thailand

Tewawong Nipaporn nipaporn.t@rsu.ac.th 1
Kowaboot Siriporn 1
Pimainog Yaowaluk 1
Watanagul Naiyana 2
Thongmee Thanunrat 3
Poovorawan Yong 3
1 Faculty of Medical Technology, Rangsit University , Muang , Pathumthani , Thailand
2 Department of Microbiology, Nopparat Rajathanee Hospital , Khannayao , Bangkok , Thailand
3 Center of Excellence in Clinical Virology, Department of Pediatrics, Faculty of Medicine, Chulalongkorn University , Pathumwan , Bangkok , Thailand
Thomas Jonathan
Electronic publication date: 2020 Dec 2
Publication date: 2020
Volume: 8
Electronic Location ID: e10453
Received 2020 Aug 19; Accepted 2020 Nov 9
Copyright: ©2020 Tewawong et al.
Copyright year: 2020
Copyright holder: Tewawong et al.
License: This is an open access article distributed under the terms of the Creative Commons Attribution License, which permits unrestricted use, distribution, reproduction and adaptation in any medium and for any purpose provided that it is properly attributed. For attribution, the original author(s), title, publication source (PeerJ) and either DOI or URL of the article must be cited.
License URL: https://creativecommons.org/licenses/by/4.0/

Keywords: Uropathogenic Escherichia coli, Phylogenetic group, Adhesin genes, Biofilm, Antimicrobial resistance

Funding: Research Institute of Rangsit University 77/2017 This work was supported by the Research Institute of Rangsit University (Grant No. 77/2017). The funders had no role in study design, data collection and analysis, decision to publish, or preparation of the manuscript.

==============================
Background

Urinary tract infections (UTIs) are the most common bacterial infections and are often caused by uropathogenic Escherichia coli (UPEC). We investigated the distribution of phylogenetic groups, adhesin genes, antimicrobial resistance, and biofilm formation in E. coli isolated from patients with UTIs.

Methods

In the present study, 208 UPEC isolated from Thai patients were classified into phylogenetic groups and adhesin genes were detected using multiplex PCR. Antimicrobial susceptibility testing was performed using agar disk diffusion. The Congo red agar method was used to determine the ability of the UPEC to form biofilm.

Results

The most prevalent UPEC strains in this study belonged to phylogenetic group B2 (58.7%), followed by group C (12.5%), group E (12.0%), and the other groups (16.8%). Among adhesin genes, the prevalence of fimH (91.8%) was highest, followed by pap (79.3%), sfa (12.0%), and afa (7.7%). The rates of resistance to fluoroquinolones, trimethoprim-sulfamethoxazole, and amoxicillin-clavulanate were  65%, 54.3%, and 36.5%, respectively. The presence of adhesin genes and antibiotic resistance were more frequent in groups B2 and C compared to the other groups. Of the 129 multidrug-resistant UPEC strains, 54% were biofilm producers. Our findings further indicated that biofilm production was significantly correlated with the pap adhesin gene (p ≤ 0.05).

Conclusion

These findings provide molecular epidemiologic data, antibiotic resistance profiles, and the potential for biofilm formation among UPEC strains that can inform further development of the appropriate prevention and control strategies for UTIs in this region.

Introduction

Urinary tract infections (UTIs) are a common bacterial infection, with 150 million UTI cases observed annually worldwide (Stamm & Norrby, 2001). Uropathogenic Escherichia coli (UPEC) is the most common causative agent of both uncomplicated and complicated UTIs, accounting for 75% and 65% of cases, respectively (Flores-Mireles et al., 2015). Clermont and colleagues developed a new polymerase chain reaction (PCR)-based method to classify the eight phylogenetic groups of E. coli, of which seven are clustered in E. coli sensu stricto (A, B1, B2, C, D, E, and F) and one belongs to Escherichia Clade 1 (Clermont et al., 2013). Several studies have reported that phylogenetic groups B2 and D are associated with extraintestinal infection, while the other groups are more prevalent among diarrheagenic and commensal bacteria (Picard et al., 1999; Kumar, Nahid & Zahra, 2017; Ahumada-Santos et al., 2020).

Adherence and colonization are the crucial steps in UTI pathogenesis. UPEC generally use various adhesins to recognize uroepithelium cells and mediate colonization (Flores-Mireles et al., 2015). Type 1 fimbriae consist of a major protein, FimA, that is associated with the ancillary proteins FimF, FimG, and the adhesin FimH, all of which are encoded by the fim gene cluster (Orndorff & Falkow, 1984). The P fimbriae are encoded by the pap gene cluster, which contains 11 genes (papA to papK) (Fernández & Berenguer, 2000). P fimbriae promote early colonization of the epithelial cells lining the tubules, while type 1 fimbriae appear to play a role in inter-bacterial binding and biofilm formation (Melican et al., 2011). The S fimbriae are expressed by the sfa operon, which was reported to be most often found in E. coli strains implicated in human meningitis and septicemia (Antao, Wieler & Ewers, 2009). The P-independent, X-binding fimbrial adhesin encoded by the afa1 operon mediates specific binding to uroepithelial cells and human erythrocyte receptors (Labigne-Roussel & Falkow, 1988). Different studies have investigated the presence of the adhesion-encoding genes pap (P fimbriae), sfa (S fimbriae), afa (afimbrial adhesin), and fimH (type 1 fimbriae) across UPEC strains using multiplex PCR (Rahdar et al., 2015; Dadi et al., 2020; Tarchouna et al., 2013; Shetty et al., 2014).

Currently, the empirical treatment of UTIs is an issue of concern due to the increasing rates of antibiotic resistance. The resistance to trimethoprim-sulfamethoxazole (TMP-SMZ), ciprofloxacin, and amoxicillin-clavulanate (AMC) among UPEC isolates is higher in developing countries (ranging from ∼50% to 85%) than in developed countries (ranging from 3% to 40%) (Kot, 2019). Routine standard antimicrobial susceptibility testing must be performed in order to reduce the rates of inappropriate empirical antibiotic therapy of UTIs and thereby decrease the occurrence of multidrug-resistant (MDR) UPEC (Adamus-Białek et al., 2018).

Biofilms are microbial communities that adhere to various surfaces, and the cells within a biofilm are encased in self-produced extracellular polymeric matrix (Hall & Mah, 2017). The ability of UPEC to form biofilms is important, as biofilms increase antimicrobial agent tolerance and facilitate evasion of the urinary tract host defense, contributing to the evolution of MDR strains and the recurrence of UTIs (Mittal, Sharma & Chaudhary, 2015).

A study of virulence genes and antimicrobial susceptibility patterns of UPEC in southern Thailand was previously reported (Themphachanal et al., 2015), but there is no information on the new classification of phylogenetic groups or the biofilm-forming ability of UPEC. Therefore, the aim of the present study was to determine the phylogenetic groups, adhesin gene distribution, antimicrobial resistance profiles, and biofilm formation ability of UPEC isolated from patients with UTIs in central Thailand. We also investigated the possible correlation between adhesin genes and the ability to form biofilm.

Materials and Methods

Ethical approval

E. coli strains were isolated from patients with UTI then identified and collected at the Nopparat Rajathanee Hospital as part of the routine microbiological laboratory. The study protocol was approved by the Ethics Review Board (ERB) of the Research Institute of Rangsit University (DPE.No.RSUERB2018-002). All the bacterial strains were acquired with permission from the Director of Nopparat Rajathanee Hospital.

Bacterial strains

The 208 non-repetitive E. coli strains isolated from urine specimens of UTI patients between February and May 2018 were used from the current study. E. coli strains were isolated from pure cultures and identified in the department of microbiological laboratory in the Nopparat Rajathanee Hospital. The bacteria were confirmed as E. coli by considering Gram’s staining morphology, colony characteristic on MacConkey agar (Oxoid, UK), and biochemical properties (Bergey & Holt, 1994). The oxidase test, catalase test, sugar fermentation, motility test, indole production, methyl red test, Voges-proskauer reaction, urease production, citrate utilization, and ornithine and lysine decarboxylase test were used as the standard biochemical testing in our laboratory. The only one isolate from each patient was investigated.

Characterization of phylogenetic groups and adhesin genes

Bacterial DNA was extracted using the optimized boiling method (Dashti et al., 2009). The phylogenetic groups of E. coli were characterized using multiplex PCR according to the protocol previously published (Clermont et al., 2013). Table S1 shows the primer sequences and the size of amplicons. In addition, four adhesin genes, pap, sfa, afa, and fimH, were detected in all isolates using multiplex PCR (Yamamoto et al., 1995; Le Bouguenec, Archambaud & Labigne, 1992; Struve & Krogfelt, 1999). The details of the primers and sizes of PCR products are listed in Table S2. The PCR reaction volume contained 15 µl of 2X AmpMaster™ HS-Taq (GeneAll®, Korea), 10 pmol/µl of each primer, 3 µl of DNA template, and DNase-free H2O to a final volume of 30 µl. Amplification was carried out in the Mastercycler® nexus (Eppendorf, Germany) under the following conditions: initial denaturation at 95 °C for 3 min, 45 cycles of 45 s denaturation at 95 °C, 45 s of primer annealing at 55 °C (to characterize the phylogenetic groups) and 54 °C (to amplify the adhesin genes), 60 s of extension at 72 °C, and further extension for 5 min at 72 °C. PCR products were separated on a 2% agarose gel with a 100-bp DNA ladder (Fermentas, US) and visualized on a UV trans-illuminator.

Antimicrobial susceptibility testing

Antimicrobial susceptibility tests were performed using the agar disk diffusion method according to Clinical and Laboratory Standards Institute guidelines (CLSI, 2018). The antibiotic disks (Oxoid, UK) ampicillin (10 µg), amoxicillin-clavulanate (20/10 µg), piperacillin-tazobactam (100/10 µg), cefoperazone-sulbactam (75/30 µg), cefazolin (30 µg), cefotaxime (30 µg), ceftriaxone (30 µg), ceftazidime (30 µg), imipenem (10 µg), meropenem (10 µg), ertapenem (10 µg), gentamicin (10 µg), amikacin (30 µg), netilmicin (30 µg), ciprofloxacin (5 µg), levofloxacin (5 µg), norfloxacin (10 µg), trimethoprim-sulfamethoxazole (1.25/23.75 µg), and fosfomycin (200 µg) were used. Escherichia coli ATCC 25922 was used as a control in all antibiogram tests. Whether a strain was MDR was determined on the basis of acquired non-susceptibility to at least one agent in three or more antimicrobial categories (Magiorakos et al., 2012).

Detection of biofilm formation

The biofilm production of all E. coli strains was determined using the Congo red agar (CRA) method, as previously published (Neupane et al., 2016; Sm, Jayakumar & Aravazhi, 2016; Tajbakhsh et al., 2016). The medium contains brain heart infusion agar (52 gm/L); sucrose (36 gm/L) and Congo red dye (0.8 gm/L). The tested organisms were cultured on CRA and incubated under the aerobic condition at 37 °C for 24 to 48 h. The six color tones of colonies were categorized as follows: very black, black, almost black, which were interpreted as strong, moderate, and weak biofilm producers, respectively, and bordeaux, red, and very red, reported as non-biofilm producers.

Statistical analysis

Chi-square test was used for comparisons of proportions the demographic characteristics of patients. The correlations between phylogenetic group, the presence of adhesin genes, biofilm production, and antimicrobial resistance were determined by performing Pearson’s chi-square tests. SPSS version 21 software was used for data analysis (IBM SPSS Inc., Armonk, NY, USA). Results were considered statistically significant if the p-value was ≤ 0.05.

Results

Among 1,926 patients with symptoms of UTI, a total of 208 isolates were identified as E. coli. The demographic characteristics of patients infected with UPEC are shown in Table 1. Among the patients, 154 (74%) were female and 54 (26%) were male. Patients were stratified into five different age groups, and those over 65 years represented 63.9% of all patients. The highest number of UPEC samples was isolated from catheter urine samples (150, 72.1%). The highest proportion of UPEC isolates came from the internal medicine ward (80, 38.5%), followed by the emergency room (45, 21.6%), intensive care unit (34, 16.3%), and outpatients (22, 10.6%).

Table 1 Demographic characteristics of patients infected with uropathogenic E. coli (N = 208).

Parameter		No. of isolates (%)	Chi-square	Degree of freedom	p-value	
Gender						
	Female	154 (74.0)	48.08a	1	<0.0001	
	Male	54 (26.0)				
Age (years)						
	<14	11 (5.3)	265.02b	4	<0.0001	
	15–24	6 (2.9)				
	25–44	15 (7.2)				
	45–64	43 (20.7)				
	≥65	133 (63.9)				
Type of samples						
	Midstream urine	58 (27.9)	40.69a	1	<0.0001	
	catheter urine	150 (72.1)				
Hospital Unit						
	Out-patient	22 (10.6)	188.23c	7	<0.0001	
	In-patient					
	Internal medicine	80 (38.5)				
	ER	45 (21.6)				
	ICU	34 (16.3)				
	Surgery	12 (5.8)				
	Pediatrics	8 (3.8)				
	Stroke	5 (2.4)				
	Burn	2 (1.0)				
MDR stains						
	MDR	129 (62.0)	12.02a	1	0.001	
	Non-MDR	79 (38.0)				
Notes.

a 0 cells (.0%) have expected frequencies less than 5. The minimum expected cell frequency is 104.0.

b 0 cells (.0%) have expected frequencies less than 5. The minimum expected cell frequency is 41.6.

c 0 cells (.0%) have expected frequencies less than 5. The minimum expected cell frequency is 26.0.

We characterized the phylogenetic groups of E. coli from urine specimens by detecting the arpA (400 bp), chuA (288 bp), yjaA (211 bp), and TspE4.C2 (152 bp) genes using multiplex PCR (Fig. 1A). Primers specific for the trpA (489 bp) gene were added to all PCR reactions to provide an internal control. Groups C and E were classified by amplification of the trpA (219 bp) and arpA (301 bp) genes using specific primers. The majority of the 208 E. coli isolates were group B2 (122, 58.7%), followed by group C (26, 12.5%), group E (25, 12%), group A (10, 4.8%), group F (9, 4.3%), group D (6, 2.9%), group B1 (5, 2.4%), unassignable (3, 1.4%), and clade I or clade II (2, 1.0%; Fig. 1B).

Figure 1 The distribution of phylogenetic groups among uropathogenic Escherichia coli isolates by the new Clermont phylo-typing method.

(A) Multiplex PCR profiles for specific uropathogenic Escherichia coli isolates by detecting the arpA (400 bp), chuA (288 bp), yjaA (211 bp), and TspE4.C2 (152 bp) genes. Lane M, 100-base pair ladder (Fermantas); Lane 1, group B2 (-, +, +, +); Lane 2, group B1 (+, -, -, +); Lane 3, group D or E (+, +, -,-); Lane 4, group B2 (-, +, +, +); Lane 5, group D or E (+, +, -,-); Lane 6, group B2 (-, +, +, +); Lane 7, group B2 (-, +, +, +); Lane 8, group B2 (-, +, +, +); Lane 9, group A or C (+, -, +, -); Lane NC, negative control. The trpA (489 bp) internal control gene appeared in all samples except the negative control. Distilled water without any DNA as negative controls was used in PCR experiments. (B) The percentage of phylogenetic groups among uropathogenic Escherichia coli isolates.

Adhesin-encoding genes were successfully amplified by multiplex PCR. The most frequent UPEC adhesin gene was fimH (191, 91.8%), followed by pap (165, 79.3%), sfa (25, 12.0%), and afa (16, 7.7%). We also investigated the adhesin gene patterns of the strains (Table 2). Among the isolates, 30 (14.4%), 167 (80.3%), and 11 (5.3%) possessed 1, 2, and 3 adhesin genes, respectively. A high prevalence of combined fimH and pap genes was significantly found (69.2%, p <  0.0001). Moreover, the fimH gene has significant association with UPEC phylogenetic groups B2 (p = 0.041). There were significant associations between phylogenetic group E and two adhesin genes namely pap and afa (p = 0.002 and p <  0.0001, respectively). Similarly, there were significant associations between phylogenetic group F and adhesin genes fimH and sfa (p = 0.005 and p = 0.044, respectively) (See Table 3).

Table 2 Profiles of adhesin genes in uropathogenic Escherichia coli strains.

No. of genes	Adhesin genes patterns	No. of isolates (%)	Chi-square	Degree of freedom	P-value	
1 gene, n = 30 (14.4%)					
	fimH	19 (9.1)				
	pap	4 (1.9)				
	sfa	5 (2.4)				
	afa	2 (1.0)				
2 genes, n = 167 (80.3%)					
	fimH, pap	144 (69.2)	922.88a	10	<0.0001	
	fimH, sfa	11 (5.3)				
	fimH, afa	6 (2.9)				
	pap, sfa	2 (1.0)				
	pap, afa	4 (1.9)				
3 genes, n = 11 (5.3%)					
	fimH, pap, sfa	7 (3.4)				
	fimH, pap, afa	4 (1.9)				
Notes.

a 0 cells (.0%) have expected frequencies less than 5. The minimum expected cell frequency is 18.9.

Table 3 The association between phylogenetic groups and adhesin genes of uropathogenic Escherichia coli isolates.

Adhesin genes	Phylogenetic group		
		B2 (n = 122)	C (n = 26)	E (n = 25)	A (n = 10)	F (n = 9)	
		B2	Non-B2	χ2	P-value	C	Non-C	χ2	P-value	E	Non-E	χ2	P-value	A	Non-A	χ2	P-value	F	Non-F	χ2	P-value	
fimH	Present	116	75	4.166	0.041	24	167	0.009	0.924	22	169	0.554	0.456	10	181	0.935	0.334	6	185	7.935	0.005	
	Absent	6	11			2	15			3	14			0	17			3	14			
pap	Present	100	65	1.254	0.263	20	145	0.105	0.746	14	151	9.428	0.002	10	155	2.738	0.098	7	158	0.014	0.907	
	Absent	22	21			6	37			11	32			0	43			2	41			
sfa	Present	19	6	3.526	0.060	1	24	1.877	0.171	1	24	1.728	0.189	0	25	1.435	0.231	3	22	4.041	0.044	
	Absent	103	80			25	158			24	159			10	173			6	177			
afa	Present	6	10	3.198	0.074	0	16	2.476	0.116	8	8	23.645	0.000	0	16	0.875	0.349	1	15	0.155	0.694	
	Absent	116	76			26	166			17	175			10	182			8	184			
Adhesin genes	Phylogenetic group		
		D (n=6)	B1 (n=5)	Unassignable (n=3)	CladeI (n=1)	Clade I or II (n=1)	
		D	Non-D	χ 2	P-value	B1	Non-B1	χ 2	P-value	Unassign	Non-unassign	χ 2	P-value	CladeI	Non-cladeI	χ 2	P-value	CladeI/II	Non-cladeI/II	χ 2	P-value	
fimH	Present	5	186	0.594	0.441	5	186	0.456	0.500	2	189	2.567	0.109	1	190	0.089	0.765	0	191	11.290	0.001	
	Absent	1	16			0	17			1	16			0	17			1	16			
pap	Present	5	160	0.060	0.806	4	161	0.001	0.970	3	162	0.739	0.373	1	164	0.262	0.609	1	164	0.262	0.609	
	Absent	1	42			1	42			0	43			0	43			0	43			
sfa	Present	0	25	0.844	0.358	1	24	0.309	0.579	0	25	0.416	0.519	0	25	0.137	0.711	0	25	0.137	0.711	
	Absent	6	177			4	179			3	180			1	182			1	182			
afa	Present	0	16	0.515	0.473	0	16	0.427	0.513	1	15	2.818	0.093	0	16	0.084	0.772	0	16	0.084	0.772	
	Absent	6	186			5	187			2	190			1	191			1	191			

We performed antimicrobial susceptibility tests on E. coli strains using different categories of antibiotics. There were significant associations between E. coli phylogenetic groups and resistance rates of antibiotics (p <  0.05) except ampicillin, gentamicin and trimethoprim-sulfamethoxazole (Table 4). All isolates showed high rates of resistance to ampicillin (84.1%), ciprofloxacin (65.4%), norfloxacin (65.4%), levofloxacin (64.9%), trimethoprim-sulfamethoxazole (54.3%), cefazolin (44.7%), cefotaxime (43.8%), ceftriaxone (43.8%), ceftazidime (43.8%), amoxicillin-clavulanate (36.5%), and gentamicin (33.7%). The rates of resistance to other antibiotics were between ∼1% and 6%. E. coli phylogenetic group C had the highest rates of resistance to all antibiotics (p <0.05) except ampicillin, gentamicin, amikacin, netilmicin, and fosfomycin (Table S3). Three isolates (1.4%) in group C were carbapenems-resistant. Interestingly, most of the 129 isolates (62.0%) that were MDR and belonged to group B2 (59.7%; 77 of 129). However, the lower resistance rates to piperacillin-tazobactam and carbapenems were observed in group B2 (p = 0.005 and p = 0.0038, respectively) (Table S3). The lowest rates of resistance to cephalosporin were observed in group A (p = 0.02), while group D was more susceptible to fluoroquinolones than the other groups (p = 0.01). The only one isolate of group A was resistant to fosfomycin (p <  0.0001).

Using the CRA method, the abilities of bacteria to form biofilm were categorized into four groups based on the color tones of colonies. Among the 95 E. coli strains that could form biofilm, 4 (4.2%) showed strong biofilm-forming ability, 38 (40.0%) showed moderate ability, and 53 (55.8%) showed weak ability. The biofilm-producing strains were predominantly clustered in phylogenetic group B2 (Table 5). Biofilm- and non-biofilm-producing UPEC showed different antimicrobial resistance profiles. Among the biofilm producers, the rate of resistance was highest for ampicillin (90%), followed by fluoroquinolones (82%), cephalosporins (50%), and gentamicin (38%). No biofilm producer was resistant to carbapenems. In contrast, the non-biofilm producers were more resistant to TMP-SMZ (58%), followed by piperacillin-tazobactam (7%) and carbapenems (3%). The frequency distribution is presented in Fig. 2. The resistance rates to ciprofloxacin, norfloxacin and levofloxacin among biofilm producers were significantly higher than non-biofilm producers (p <  0.0001; Fig. 2). Of the 129 MDR E. coli isolates, 54% were biofilm producers.

Table 4 Chi-square test for comparisons of resistance rates to antimicrobial agents among various phylogenetic groups of uropathogenic Escherichia coli isolates.

Antimicrobial resistance rates	Phylogenetic group	Chi- square	P-value	
	B2	C	E	A	F	D	B1	Unassignable	Clade I and I or II	Total			
	n = 122(%)	n = 26(%)	n = 25(%)	n = 10(%)	n = 9(%)	n = 6(%)	n = 5(%)	n = 3(%)	n = 2(%)	n = 208(%)			
Penicillins	
AMP	101 (82.8)	25 (96.2)	21 (84)	8 (80)	6 (66.7)	6 (100)	4 (80)	3 (100)	1 (50)	175 (84.1)	16.707	0.054	
β-lactam/ β-lactamase inhibitor combinations	
AMC	39 (32)	17 (65.4)	9 (36)	3 (30)	4 (44.4)	1 (16.7)	3 (60)	0	0	76 (36.5)	16.906	0.050	
TZP	2 (1.6)	7 (26.9)	1 (4)	0	1 (11.1)	0	0	0	0	11 (5.3)	29.961	0.000	
SCF	5 (4.1)	7 (26.9)	1 (4)	0	0	0	0	0	0	13 (6.3)	22.477	0.007	
Cephalosporins	
KZ	54 (44.3)	18 (69.2)	11 (44)	1 (10)	5 (55.6)	2 (33.3)	1 (20)	1 (33.3)	0	93 (44.7)	15.248	0.084	
CTX	53 (43.4)	18 (69.2)	11 (44)	1 (10)	5 (55.6)	1 (16.7)	1 (20)	1 (33.3)	0	91 (43.8)	16.977	0.049	
CRO	53 (43.4)	18 (69.2)	11 (44)	1 (10)	5 (55.6)	1 (16.7)	1 (20)	1 (33.3)	0	91 (43.8)	16.977	0.049	
CAZ	52 (42.6)	18 (69.2)	11 (44)	1 (10)	5 (55.6)	1 (16.7)	2 (40)	1 (33.3)	0	91 (43.8)	16.977	0.049	
Carbapenems	
IPM	0	3 (11.5)	0	0	0	0	0	0	0	3 (1.4)	21.307	0.011	
MEM	0	3 (11.5)	0	0	0	0	0	0	0	3 (1.4)	21.307	0.011	
ERT	0	3 (11.5)	0	0	0	0	0	0	0	3 (1.4)	21.307	0.011	
Aminoglycosides	
CN	43 (33.6)	12 (46.2)	9 (36)	1(10)	4 (44.4)	1 (16.7)	1 (20)	0	0	70 (33.7)	10.759	0.293	
AK	0	0	0	0	1 (11.1)	0	0	0	0	1 (0.5)	25.121	0.003	
NET	0	0	0	0	1 (11.1)	0	0	0	0	1 (0.5)	25.121	0.003	
Fluoroquinolones	
CIP	87 (71.3)	25 (96.2)	10 (40)	4 (40)	5 (55.6)	1 (16.7)	2 (40)	2 (66.7)	0	136 (65.4)	36.148	0.000	
NOR	88 (72.1)	25 (96.2)	10 (40)	4 (40)	5 (55.6)	1 (16.7)	1 (20)	2 (66.7)	0	136 (65.4)	36.148	0.000	
LEV	86 (70.5)	25 (96.2)	10 (40)	4 (40)	5 (55.6)	1 (16.7)	2 (40)	2 (66.7)	0	135 (64.9)	35.411	0.000	
Folate pathway inhibitors	
SXT	61 (50)	20 (76.9)	16 (64)	5 (50)	5 (55.6)	4 (66.7)	0	1 (33.3)	0	113 (54.3)	10.853	0.286	
Fosfomycins	
FOS	0	0	0	1 (10)	0	0	0	0	0	1 (0.5)	19.896	0.019	
Notes.

Amp ampicillin

AMC amoxicillin-clavulanic acid

TZP piperacillin-tazobactam

SCF cefoperazone-sulbactam

KZ cefazolin

CTX cefotaxime

CRO ceftriaxone

CAZ ceftazidime

CN gentamicin

CIP ciprofloxacin

NOR norfloxacin

LEV levofloxacin

SXT trimethoprim-sulfamethoxazole

IPM Imipenem

MEM meropenem

ERT ertapenem

CN gentamicin

AK amikacin

NET netilmicin

FOS fosfomycin

Table 5 Biofilm forming ability among various phylogenetic groups of uropathogenic Escherichia coli isolates.

Phylogenetic group	Prevalence of biofilm formation ability	
	Strong (n = 4), %	Moderate (n = 38), %	Weak (n = 53), %	Absent (n = 113), %	
B2 (n = 122)	3 (2.5)	36 (29.5)	46 (37.7)	37 (30.3)	
C (n = 26)	0	0	3 (11.5)	23 (88.5)	
E (n = 25)	0	0	1 (4)	24 (96)	
A (n = 10)	0	0	0	10 (100)	
F (n = 9)	0	1 (11.1)	1 (11.1)	7 (77.8)	
D (n = 6)	0	0	0	6 (100)	
B1 (n = 5)	1 (20)	0	1 (20)	3 (60)	
Unassignable (n = 3)	0	1 (33.3)	0	2 (66.7)	
Clade I (n = 1)	0	0	0	1 (100)	
Clade I or II (n = 1)	0	0	1 (100)	0	

Figure 2 Comparison of antibiotic resistance between biofilm producers and non-biofilm producers.

Uropathogenic E. coli strains were evaluated for in vitro susceptibility to nineteen antibiotics: Amp, ampicillin; AMC, amoxicillin-clavulanic acid; TZP, piperacillin-tazobactam; SCF, cefoperazone-sulbactam; KZ, cefazolin; CTX, cefotaxime; CRO, ceftriaxone; CAZ, ceftazidime; CN, gentamicin; CIP, ciprofloxacin; NOR, norfloxacin; LEV, levofloxacin; SXT, trimethoprim-sulfamethoxazole; IPM, Imipenem; MEM, meropenem; ERT, ertapenem; CN, gentamicin; AK, amikacin; NET, netilmicin; FOS, fosfomycin. Bar graphs show the percentage of antibiotic resistance among biofilm producers in blue and non-biofilm producers in orange.

We also investigated the association between the presence or absence of the four adhesin genes and biofilm formation ability. The results demonstrated that biofilm production was significantly correlated with the presence of pap adhesin gene (p ≤ 0.05; Table 6). Among the biofilm producer group, we found the prevalence of pap gene was lower in strong biofilm formers than in weak and moderate.

Table 6 Prevalence of virulence genes among various groups of different biofilm formation ability.

Virulence	Percentage of biofilm formation ability			
genes	Strong
(n = 4), %	Moderate
(n = 38), %	Weak
(n = 53), %	Total
(n = 95), %	Absent
(n = 113), %	Pearson
Chi-square	p-value	
fimH	4 (100)	37 (97.4)	49 (92.5)	90 (94.7)	101 (89.4)	1.97a	0.16	
pap	2 (50)	35 (92.1)	44 (83.0)	81 (85.3)	84 (74.3)	3.76b	0.05	
sfa	0	1 (2.6)	6 (11.3)	7 (7.4)	18 (15.9)	3.58c	0.06	
afa	0	2 (5.3)	3 (5.7)	5 (5.3)	11 (9.7)	1.45d	0.23	
Notes.

P-values were calculated using the Pearson Chi-squared test. P-values ≤ 0.05 are indicated in bold.

a 0 cells (.0%) have expected count less than 5. The minimum expected count is 7.76.

b 0 cells (.0%) have expected count less than 5. The minimum expected count is 19.64.

c 0 cells (.0%) have expected count less than 5. The minimum expected count is 11.42.

d 0 cells (.0%) have expected count less than 5. The minimum expected count is 7.31.

Discussion

The higher proportion of UTIs in female (74%) than male (26%) patients in this study was observed. This is most likely to the anatomical structure of the female urethra, which is shorter, wider, and closer to the anus than that of males. E. coli is common in the gastrointestinal tract flora and can be easily moved from the anus to the urinary tract, leading to UTIs (Dadi et al., 2020). Half of the UTI cases in this study (50%) were observed in female patients over 65 years of age. In postmenopausal women, the low level of estrogen and high intravaginal pH are associated with increased bacterial adherence to the uroepithelium cell, which causes UTIs (Johansson et al., 1996; Beyer et al., 2001). Our study included a large number of catheter urine specimens, which was correlated with the high percentage of infections in the over-65 age group. The low immunity level in the elderly puts those of advanced age at a high risk of bacterial infection and is responsible for the high prevalence in catheterized cases (Themphachanal et al., 2015).

Phylogenetic groups B2 and D are common strains implicated in UTIs (Ejrnæs et al., 2011). In contrast to the results of studies from Uruguay and Southern Thailand, where high prevalences of phylogenetic group D were found (Themphachanal et al., 2015; Robino et al., 2014), we observed that group B2 was the most prevalent UPEC (58.7%), followed by group C (12.5%). Our results are in accordance with several studies in which the dominant strain was found to be group B2. These studies were conducted in North America (45% prevalence of group B2) (Johnson et al., 2003), Denmark (67%) (Ejrnæs et al., 2011), Poland (35%) (Kot et al., 2016), South Korea (79%) (Lee et al., 2016), and Ethiopia (30%) (Dadi et al., 2020). Using a novel PCR-based method (Clermont et al., 2013), we could classify UPEC into groups C, E, and F and clade I, resulting in a lower percentage of strains in groups A, B1, and D than in earlier studies. This finding indicates that the triplex method of phylo-grouping misidentifies groups C, E, and F and clade I as belonging to group A, B1, B2, or D (Kumar, Nahid & Zahra, 2017). It had been reveal that some strains (1.4%) could not be assigned to a phylogenetic group due to simply relying upon PCR of a few small number of genes. As stated by Clermont et al. (2013), the unassignable strains are more likely the result of large-scale recombination events from two different groups or genome plasticity driven by loss and gain of genes. In this study, 1% of UPEC belonged to cryptic clade I/II. This is a much lower percentage than in a study conducted in Mexico (9%) (Kumar, Nahid & Zahra, 2017). The cryptic clades are primarily associated with environmental E. coli; thus, the observed results may be related to a lack of good hygiene practices. The different distributions of phylogenetic groups may depend on the geographic area, health status of the host, use of antibiotics, and/or variations in research design and sample size of the studies (Derakhshandeh et al., 2013).

The most prevalent adhesin gene was fimH, followed by pap, sfa, and afa. In agreement with studies conducted in Ethiopia (Dadi et al., 2020) and Iran (Tajbakhsh et al., 2016), phylogenetic group B2 strains showed the highest frequency of the adhesin genes in our study. We found a coexistence of fimH and pap genes (69.2%), indicating a high presence of virulence genes among UPEC isolated from UTI patients in Thailand. This outcome was different from that of a study in Iran, in which the combination of pap and afa virulence genes was more common (Rahdar et al., 2015). The ability of UPEC to form biofilm is a crucial virulence property. We found that 45.7% of UPEC were biofilm producers and that most of these classified into phylogenetic group B2. This finding demonstrates that biofilm formation may be associated with phylogenetic group B2. The association between biofilm-forming ability and some adhesin genes among UPEC was previously reported (Rahdar et al., 2015; Tajbakhsh et al., 2016; Naves et al., 2008). Consistently, the most significant correlation observed in our study was the correlation between the pap gene and biofilm production. The negative correlation found closely to significance between sfa gene and biofilm formation ( p = 0.06), as the prevalence of this gene was lower in biofilm producer. In contrast, no significant correlation was seen between the fimH, or afa genes and biofilm production in the strains evaluated in this study. This finding is in agreement with other studies that did not find significant correlations in clinical isolates of pathogenic E. coli (Reisner et al., 2006; Hancock, Ferrie‘res & Klemm, 2007). The discrepant results imply that these genes are not the only determinants of biofilm production in UPEC strains; rather, environmental and genetic factors may also be involved (Reisner et al., 2006). Adhesin genes such as fimH are under strict control by phase variation in many strains. The presence of adhesin genes certainly does not imply their expression. It would have been far more informative if the further study had been able to correlate expression of these genes rather than just their presence or absence by PCR.

It is important to perform antimicrobial susceptibility testing to select the appropriate empiric antibiotic therapy for UTIs. Our findings showed that the rate of resistance to ampicillin (84.1%) was higher than rates of resistance to other antibiotics. In general, fluoroquinolones are recommended for oral antimicrobial therapy in uncomplicated pyelonephritis. TMP-SMZ is commonly used in the treatment of uncomplicated cystitis, while AMC was a first-line therapy for complicated UTIs (Bonkat et al., 2019). However, our results revealed that rates of resistance to fluoroquinolones, TMP-SMZ, and AMC were 65%, 54%, and 37%, respectively. This result is consistent with a previous mini-review reporting increases in resistance rates of those drugs among UPEC isolates in developing countries (Kot, 2019). This likely emerged due to the widespread use of fluoroquinolones for uncomplicated UTIs or the inappropriate use of TMP-SMZ for empiric UTI treatment (Bartoletti et al., 2016). In this study, the strains in phylogenetic group C showed the highest rates of antibiotic resistance. In recent decades, the increasing rate of MDR in UPEC has become a public health threat. A high prevalence of MDR UPEC of approximately 62% was observed in the current study, similar to the findings reported in Iran (60.2%) (Tajbakhsh et al., 2016) and Nepal (63.2%) (Ganesh et al., 2019). The majority of MDR UPEC belonged to phylogenetic group B2, consistent with the outcomes reported in South Korea (73%) (Lee et al., 2016).

The present study found that biofilm producer strains were more resistant to ciprofloxacin, norfloxacin and levofloxacin than non-biofilm producers. These results were in agreement with previous studies indicating that the sessile bacterial cells are much less susceptible to antimicrobial agents than nonattached (planktonic) cells (Costerton, Stewart & Greenberg, 1999). A higher rate of resistance to TMP-SMZ was found among the non-biofilm producers than among the biofilm producers. One explanation for this finding is that these strains may carry the dhfr or dhps gene mutation on chromosomal DNA, which are common causes of resistance to this drug (Huovinen et al., 1995).

In conclusion, the majority of UPEC among patients with UTIs in this geographical area belonged to phylogenetic group B2. UPEC in this group also showed the highest prevalence of adhesin genes and biofilm formation. The analysis of the antimicrobial resistance of strains tested in this study showed a high level of resistance to cephalosporins, fluoroquinolones, TMP-SMZ, and AMC among strains belonging to groups B2 and C. Therefore, further study of the molecular epidemiology of UPEC and their antibiotic susceptibility patterns will improve our understanding of the organism and lead to a better management of UTIs.

Supplemental Information

Supplemental Information 1 Sequence of oligonucleotide primers used for detection of the phylogenetic groups

Click here for additional data file.

Supplemental Information 2 Sequence of oligonucleotide primers used for amplification of the adhesin genes

Click here for additional data file.

Supplemental Information 3 The association between phylogenetic group and antimicrobial susceptibility patterns of uropathogenic Escherichia coli isolates

Click here for additional data file.

Supplemental Information 4 Multiplex PCR profiles for specific uropathogenic Escherichia coli isolates according to the new Clermont phylo-typing method (Uncropped)

Click here for additional data file.

Supplemental Information 5 The biofilm ability of uropathogenic Escherichia coli were determined by Congo Red Agar

R, red; AB, almost black; VB, very black. The color tones of colonies were categorized as follows: very black, and almost black, which were interpreted as strong, and weak biofilm producers, respectively, and red reported as non-biofilm producers.

Click here for additional data file.

Supplemental Information 6 Patient data and raw data for statistical analysis

Click here for additional data file.

We would like to acknowledge the staff of the Center of Excellence in Clinical Virology, Faculty of Medicine, Chulalongkorn University, for their excellent technical assistance. We also thank all of the medical technicians in the hospital for helping with bacteria collection and Ms. Naraumon Beakee, Ms. Pacharida Pattum, Ms. Sakonwan Thanoochan, and Ms. Thanyaporn Sidafong for assistance with the laboratory process.

Additional Information and Declarations

Competing Interests

Author Contributions

Ethics

Data Availability

The authors declare there are no competing interests.

Nipaporn Tewawong conceived and designed the experiments, performed the experiments, analyzed the data, prepared figures and/or tables, authored or reviewed drafts of the paper, and approved the final draft.

Siriporn Kowaboot analyzed the data, prepared figures and/or tables, and approved the final draft.

Yaowaluk Pimainog and Yong Poovorawan conceived and designed the experiments, authored or reviewed drafts of the paper, and approved the final draft.

Naiyana Watanagul and Thanunrat Thongmee performed the experiments, prepared figures and/or tables, and approved the final draft.

The following information was supplied relating to ethical approvals (i.e., approving body and any reference numbers):

The study protocol was approved by the Ethics Review Board (ERB) of the Research Institute of Rangsit University (DPE.No. RSUERB2018-002).

The following information was supplied regarding data availability:

Raw data are available in the Supplementary Files.

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
