# Peer review of "Distribution of phylogenetic groups, adhesin genes, biofilm formation, and antimicrobial resistance of uropathogenic Escherichia coli isolated from hospitalized patients in Thailand"

_PeerJ, doi:10.7717/peerj.10453_

## Round 0.1 · original submission · Minor Revisions

Both reviewers were generally positive about the manuscript, although both have expressed concern about aspects of the statistical analyses, so particular attention should be paid to this part of the paper.

Reviewer 1 ·

Basic reporting

In this paper, Tewawong et al. study the phylogenic grouping, antimicrobial resistance, adhesin gene presence and biofilm-forming capability of 208 Thai UPEC isolates. The work contains a considerable amount of information that will support UTI research in this area. The authors note in the Introduction that there has been a previous study on UPEC in Thailand but that this study provides novel data on phylogenic groups and biofilms.

The paper is well-structured and very well-referenced. Sufficient context for all aspects of the work is provided in the Introduction. The paper is self-contained and the authors write in clear and appropriate English throughout. Ethical approval was granted for the study. I thank the authors for including the raw dataset and uncropped images with this manuscript.

Minor typographical issues:

Line 58: There should be a comma after FimA.

Line 126: Should “cefoperazone/sulbactam” be written as “cefoperazone-sulbactam” for consistency with the other dual antibiotic discs used?

Line 238: Missing word – “This finding demonstrates that biofilm formation may be associated with [WHICH?] phylogenetic group.”

Fig 1B: “The pie chart show” does not need to be stated in the legend.

Table 4: Percentage scores are given in the table in parenthesis, therefore column headers should preferably be written “n = X (%)” rather than “n = X, %”.

Experimental design

A reasonable method of selection for the E. coli isolates is given, though there should be more detail on the confirmation methodology, given how closely-related E. coli is to some other species and the variety of biochemical differences between E. coli strains and pathovars.

Full details of primers and PCR conditions, antibiotic testing and Congo red biofilm analyses are provided. One aspect of concern is that the paper cited for Congo red biofilm assay relates to a Gram positive organism (Staphylococcus epidermidis) and not a Gram negative organism such as E. coli. Since Congo red reacts with various Gram negative cell-surface structures (e.g. Curli fimbriae, PMID: 26485271; type III secretion systems, PMID: 1802051) and is used as a specific assay for those molecules, I am not certain whether this methodology has been appropriately applied.

Issues to be addressed:

Lines 103-104: Please give brief details of the “standard biochemical tests” used to confirm the isolates were E. coli.

Lines 134-139: As noted above, I have concerns regarding the use of a Staphylococcus assay to test for biofilm formation in E. coli. The authors should comment on how this protocol was adjusted or adapted to fit the present study (or at the very least, whether this was considered at all). It would be preferable if the authors could show images of controls, e.g. well-characterised high- and no-biofilm producing strains, or cite a previous E. coli study. See also comments on Fig S2, below.

Lines 140-143: Data for statistical tests are well-tabulated in the Results section but there is no mention of adjusting for multiple comparisons. Was this carried out?

Fig 1A: The authors should state what negative control was used, e.g. water only or DNA from an unrelated species.

Fig S2: There is a typographical error (“very blank” rather than “very black”) on the image. Moreover, the rest of the segments are labelled with their biofilm properties (e.g. weak/none) rather than their colour – please make this consistent. The figure legend and method description in lines 137-139 suggest that “very black” should be “weak” biofilm formation, which does not correlate with the labelling on the image, where the “weak” labels seem to be on “almost-black” streaks. The authors should change the order of the biofilm descriptors so that they match the order of the colours in both the legend and methods section.

Validity of the findings

In general, I find the authors’ conclusions are reasonable given the data, despite my concerns regarding the biofilm assay. The Discussion is broad and takes into account a significant amount of previous work in the field, particularly when drawing links to past PCR-based studies. There are some minor issues as outlined below.

Lines 197-198: The authors explain the sex imbalance in the study using reasonable biological explanations, but since these explanations were not actually tested during the study, such a statement should be qualified by saying “most likely” or similar.

Lines 220-222: The authors state that three strains could not be assigned to phylogenetic groups “due to large-scale recombination events from two different groups or to genome plasticity driven by loss and gain of genes “. Since the authors did not perform an analysis that would enable them to make this claim and are simply relying upon PCR of a few small number of genes, their statement should again be speculative, as above.

Lines 241-243: The authors have stated that their p value cutoff is 0.05, so it is reasonable for them to make this statement, however it would be appropriate for the authors to mention their sfa result (p = 0.06) here, given how close it is to significance.

Line 247: Adhesin genes such as fimH are under strict control by phase variation in many strains (and some are “locked on” or “locked off” through mutation). The authors should note in their Discussion that the presence of adhesin genes certainly does not imply their expression. It would have been far more informative if the authors had been able to correlate expression of these genes rather than just their presence/absence by PCR – so a comment on this in the Discussion seems appropriate.

Lines 263-264: I’m not sure what claim the authors are making here – could this be rephrased?

Lines 265-266: Did the authors perform a statistical analysis to back up this claim? If so, please cite the relevant p values in the Results section – otherwise consider rephrasing this statement.

Additional comments

n/a

Reviewer 2 ·

Basic reporting

The manuscript is well written, language is professional and easy to follow, with enough background information that make the big picture of this work very clear. A sentence in the end of introduction that evidences the main question of this work and how it fills a knowledge gap in the field be appreciated.

Experimental design

The material and methods are very well described with sufficient detail or referenced when appropriate.

Validity of the findings

This work analyzed a large collection of UPEC samples for phylogenetic groups, presence of adhesins, biofilm formation and antimicrobial resistance and it provides interesting results that were analyzed in a very clear and organized way. If there is a weakness, is that it misses statistical analysis (as noted below in general comments for the author), to show significance of the findings and give the work a higher impact.

Additional comments

Specific comments for Results section

Please mention table 2 in results section
Was the association between phylogenetic groups and adhesins statistically significant? It is not clear if statistical analysis was performed in this case? IF not, it would be interested to see if these associations were significant. Same applies to the relationship between antimicrobial resistance genes and phylogenetic groups.
The correlation found between pap adhesin gene and biofilm formation seems to be negative in this case, as the prevalence of pap was lower in strong biofilm formers than in weak and moderate. Please make it clear in the text (lines 192-194)
Table 1: OK
Table 2: not mentioned in the text (line 166); missing stats analysis
Table 3: missing stats analysis
Table 4: missing stats analysis
Table 5: OK
Table 6: OK
Figure 1: OK
Figure 2: OK. Seems like there could be a significant difference in resistance to ciprofloxacin, norfloxacin and levofloxacin between biofilm producer and non producer, but there is no statistical analysis to support it.
Supplemental material
Table S1 and S2: please fix title of table to start with “Sequence of oligonucleotides…” or “Sequence of primers…”.
Figure S2: according to the results section, “very black” correspond to weak biofilm. However, in this figure you have “very black” and “weak” as different things. Please clarify.

---

## Round 0.2 · accepted · Accept

Both reviewers were happy that all scientific concerns had been addressed. Well done.

Reviewer 1 ·

Basic reporting

The authors have revised the manuscript as recommended and I have no further scientific concerns. There are some minor issues with English that have crept in as a result of the modifications, so the authors may wish to double-check the wording of the revised text. For example, the use of plural vs singular words is sometimes incorrect (e.g. line 187: "association" should be "associations"; line 142: "organism were" should be "organisms were").

Experimental design

The authors have clarified aspects of their experimental design that I queried previously. The manuscript has been much improved by these additional details.

Validity of the findings

The authors have revised their manuscript's Discussion appropriately; in particular, they have noted where their conclusions are drawn from the wider literature as opposed to their own results. Again, this improves the manuscript considerably.

Reviewer 2 ·

Basic reporting

No additional comments.

Experimental design

No additional comments.

Validity of the findings

No additional comments.